# Hydroxyapatite Coating on Ti-6Al-7Nb Alloy by Plasma Electrolytic Oxidation in Salt-Based Electrolyte

**DOI:** 10.3390/ma15207374

**Published:** 2022-10-21

**Authors:** Avital Schwartz, Alexey Kossenko, Michael Zinigrad, Yosef Gofer, Konstantin Borodianskiy, Alexander Sobolev

**Affiliations:** 1Department of Chemical Engineering, Ariel University, Ariel 4070000, Israel; 2Department of Chemistry, Bar Ilan University, Ramat-Gan 5290002, Israel

**Keywords:** biomaterials, coatings, corrosion resistance, hydroxyapatite, medical applications, molten salt, plasma electrolytic oxidation, titanium dioxide

## Abstract

Titanium alloys have good biocompatibility and good mechanical properties, making them particularly suitable for dental and orthopedic implants. Improving their osseointegration with human bones is one of the most essential tasks. This can be achieved by developing hydroxyapatite (HA) on the treating surface using the plasma electrolytic oxidation (PEO) method in molten salt. In this study, a coating of titanium oxide-containing HA nanoparticles was formed on Ti-6Al-7Nb alloy by PEO in molten salt. Then, samples were subjected to hydrothermal treatment (HTT) to form HA crystals sized 0.5 to 1 μm. The effect of the current and voltage frequency for the creation of the coating on the morphology, chemical, and phase composition was studied. The anti-corrosion properties of the samples were studied using the potentiodynamic polarization test (PPT) and electrochemical impedance spectroscopy (EIS). An assessment of the morphology of the sample formed at a frequency of 100 Hz shows that the structure of this coating has a uniform submicron porosity, and its surface shows high hydrophilicity and anti-corrosion properties (4.90 × 10^6^ Ohm·cm^2^). In this work, for the first time, the process of formation of a bioactive coating consisting of titanium oxides and HA was studied by the PEO method in molten salts.

## 1. Introduction

Metal and polymer artificial implant materials are wieldy employed for the repair and reconstruction of damaged bones. Among the metal bioimplants, titanium (Ti) and its alloys are mostly used for load-bearing parts [1,2,3]. Particularly, Ti-6Al-4V is the most applicable and commercially available alloy for these applications because of its better mechanical (biotribology, biomechanics) properties and good biocompatibility [4,5,6]. However, a study by J.L. Woodman et al. [7] showed that this alloy can release vanadium ions [8], which have a high cytotoxic effect and other adverse effects [9]. In general, this type of material must have several requirements: Low cytotoxicity, high corrosion and mechanical properties, and good bioactivity and wettability for the osseointegration process [10,11,12,13]. Recently, several binary and trinary titanium alloys free of vanadium and other toxic elements [14] were developed to solve the abovementioned requirements. The most promising material is Ti-6Al-7Nb, which produces a bioactive “natural” coating [15] on a surface during the osseointegration process. However, its insufficient bioactivity usually disrupts the formation of strong bonds with human tissue (bones), reducing its application in the human body [16]. Therefore, several research works in the past decade have focused on the surface modification of implants to obtain a bioactive coating based on TiO_2_ [17]. Research has shown that hydroxyapatite (HA) can lead to better results, since the material resembles natural bone tissue in physical and chemical properties [18,19]. The use of an HA coating provides a significant increase in the contact area between the implant and bone, which leads to a noticeable improvement in the short-term indication for viability compared to “raw” titanium implants [20]. It has been shown that an HA coating [21] increases the quality of bone tissue coalescence, leads to rigidity fixation for implanted structures, and stimulates the growth of mature bone tissue. Simultaneously, it has been established that the chemical composition and morphology of the surface are essential characteristics to determine the body’s reaction to foreign material [22].

Hence, several methods are used to synthesize HA coatings, including the sol-gel method [23], electrophoretic and electrochemical deposition [24,25], microwave sintering [26], 3D printing [27], magnetron sputtering [28], and others. However, most of these methods are associated with high process temperatures, which affects the microstructure of the implant material and, consequently, its bioactivity and biocompatibility. Furthermore, additional treatments and multistage processes are used to increase the adhesion of the coating to the substrate [29], which increases the total cost of the implants and does not cause the desired effect.

In recent years, the plasma electrolytic oxidation (PEO) method [30] has been used as an alternative approach to the production of ceramic coatings with high adhesion and anticorrosion properties [31]. The PEO method is based on anodic or alternating current polarization of the processed material in aqueous electrolytes at a high voltage, producing micro arc discharges that flow on the electrode surface [32,33]. As a result of the local high-energy impact, coatings that form on a metallic surface include elements of the metallic matrix (oxidized metal) and elements of the electrolyte [34,35,36,37]. The bioactive coating fabricated by PEO methods in an aqueous solution usually consists of rutile and anatase crystalline phases with a small amount of HA. However, the introduced HA has low crystallinity; therefore, the process must be accompanied by an additional hydrothermal treatment [38,39,40]. Amorphous apatite undergoes a phase transition during hydrothermal treatment, turning into HA. The resulting crystals increase the adsorption-chemical interaction [41] of physiological fluids (calcium and phosphorus ions) with the implant surface, thereby positively affecting the bioactivity and rate of osseointegration. 

There are several main drawbacks in the PEO process in an aqueous electrolyte, namely, the use of high currents and voltages, which is decisive, since the dimensions of the processed samples are limited by the power supplied from the power source. In addition, the PEO process in aqueous electrolytes has several disadvantages, such as electrolyte heating, high current density, and the appearance of undesirable impurities in the newly formed coating due to electrolyte decomposition. It should also be noted that the coating has a two-layer porous structure and contains impurities from the composition of the electrolyte, which worsens the mechanical, anticorrosive, and cytotoxic properties of the coating. In addition, the formed HA crystals have a coarse-grained structure, which negatively affects their mechanical and tribological properties and often leads to their destruction during implantation. Our previous studies have shown that these problems can be solved by conducting a PEO process in molten salt. We studied and analyzed the coating surface and its corrosion properties obtained for Ti-6Al-4V [42] and Al1050 alloys [43] using PEO in molten salt.

In this work, for the first time, the process of formation of a bioactive coating consisting of titanium oxides and HA is studied using the PEO method in molten nitrate salts, and we also study the influence of the PEO current and voltage frequency on the morphology, chemical and phase composition, wettability, and corrosion properties of fabricated coatings on a Ti-6Al-7Nb alloy. The morphology, phase, and chemical composition of the coatings were examined using scanning electron microscopy (SEM), X-ray diffraction (XRD), and X-ray photoelectron spectroscopy (XPS), respectively. The wettability properties of the surface were investigated using Hanks’ solution by the sessile drop method with contact angle measurements. Finally, the corrosion behavior of the developed coatings was studied using PPT and EIS tests in Hanks’ solution.

The coating obtained in molten salts does not contain undesirable impurities from the electrolyte composition. It has a two-layer coating structure with an inner dense layer and an outer porous layer with evenly distributed small HA crystals that have high mechanical and corrosion resistance with good bioactive and osseointegration properties.

## 2. Materials and Methods

### 2.1. Samples and Electrolytes

Titanium alloy (Ti-6Al-7Nb) plates (Scope Metals Group Ltd., Bne Ayish, Israel) with a surface area of 16 cm^2^ were prepared as substrates for the PEO process. The chemical composition of the untreated samples is shown in Table 1.

Samples were successively polished with sandpaper with a grain size of 240–1200, followed by washing with acetone, ethanol, and distilled water. These samples were treated by a two-stage procedure. In the first stage, the PEO method was applied to obtain a basic oxide coating. An oxide-ceramic layer consisting of titanium oxides (rutile and anatase) and bioactive HA species was formed. In the second stage, the samples were subjected to a hydrothermal treatment reactor to obtain HA crystals from the finely dispersed powdered HA that was introduced earlier.

The eutectic mixture NaNO_3_–KNO_3_ (Sigma–Aldrich Co., St. Louis, MO, USA) with the addition of 1 wt.% fine (40–60 nm) apatite crystalline powder (Sigma–Aldrich Co., St. Louis, MO, USA) at a temperature of 280 °C was used as an electrolyte for PEO. A nickel crucible (Scope Metals Group Ltd., Israel) served as a bath and counter electrode. PEO processing was carried out using an MP2-AS 35 switching power supply (Magpulls, Sinzheim, Germany) at a current density of 0.045 A/cm^2^ in unipolar mode (UP+) and a square wave signal frequency of 100–500 Hz with a duty cycle of 50%. The electrical parameters for the process were recorded using a Fluke Scope-Meter 199C oscilloscope (200 MHz/2.5 GS s^−1^, Fluke, Everett, WA, USA). The following designation for the samples was adopted in this work: S0—untreated sample; S1—sample processed at a frequency of 100 Hz; S2—sample processed at a frequency of 300 Hz; and S3—sample processed at a frequency of 500 Hz. After PEO treatment, the samples were washed successively with distilled water and ethanol followed by drying.

The obtained samples were treated in a Berghof BR–100 hydrothermal treatment reactor (Berghof GmbH, Eningen unter Achalm, Germany) in a KOH solution (Sigma–Aldrich Co., St. Louis, MO, USA) (pH = 11) at a temperature of 200 °C for 2 h. The resulting samples were designated as S1 HTT–S3 HTT.

### 2.2. Surface Analysis

The phase composition of the PEO coatings was analyzed by XRD (SmartLab SE, Rigaku, Tokyo, Japan) using Cu Kα = 0.154 nm radiation at 40 kV and 30 mA in the angular range of 20 to 80° (grazing incident 2-theta configuration with an incident angle of 2°) with 0.03°/step at a scanning speed of 0.3°/min. The acquisition of the diffraction pattern and phase recognition was carried out using the program SmartLab Studio II v.4.2.44.0 and the PDF-2 2019 ICDD diffraction database.

The surface morphology and cross-section of the coatings were studied using a TESCAN MAIA3 scanning electron microscope (SEM, Brno, Czech Republic). The elemental composition was determined using energy dispersive spectroscopy (EDS, Oxford Instruments, Abingdon, UK) with an X-MaxN detector.

X-ray photoelectron spectroscopy (XPS) analysis was carried out using a Nexsa G2 spectrometer (Thermo Fisher Scientific, Oxford, UK) equipped with a monochromated Al Kα X-ray radiation source (photon energy 1486.6 eV, 72 W) under ultrahigh vacuum conditions (base pressure, 9.6 × 10^−9^ mbar).

Survey spectra were acquired at pass energies of 200 eV and 50 eV, respectively. The beam diameter was 400 mm. All measurements were carried out with a combination of a low-energy electron shower gun and a low-energy Ar-ions shower for charge neutralization. All spectra were measured on as-is samples without prior manipulation of sputtering.

The contact angle (CA) was measured using a homemade goniometer in sessile-drop mode on an RH-2000-3D digital microscopy system (Hirox Ltd., Tokyo, Japan) with a 6 µL volume drop of Hanks’ solution (Sigma–Aldrich Co., USA). The mean CA and standard deviation were calculated from measurements of five drops on each surface.

The surface porosity of the samples after the PEO process was analyzed using the ImageJ ver. 1.53m (National Institutes of Health, Bethesda, MD, USA) software based on the surface morphologies obtained by SEM images.

### 2.3. Electrochemical Methods

Corrosion and electrochemical tests were carried out in potentiodynamic mode using impedance spectroscopy using a PARSTAT 4000A potentiostat/galvanostat (Princeton Applied Research, Oak Ridge, TN, USA) in Hanks’ solution (Sigma–Aldrich Co., USA) utilizing a 100 mL three-electrode cell (Figure 1). 

The test samples (S0, S1 HTT–S3 HTT) served as the working electrode. A Pt mesh (Sigma–Aldrich Co., USA) was used as a counter electrode, and Ag/AgCl 3.5 M KCl was used as a reference electrode (Metrohm Autolab B.V., Utrecht, The Netherlands). Three tests were carried out on each sample; the mean and standard deviation were calculated and presented along with the results.

Before testing, the test samples were kept for an hour in Hanks’ solution to establish the open-circuit potential (OCP). Potentiodynamic polarization curves were recorded in the potential range of ±250 mV relative to the OCP at a scanning rate of 0.1 mV/s.

EIS data were acquired at the open-circuit potential (OCP) over a wide frequency range from 100 kHz to 1 mHz using a sinusoidal voltage amplitude of 5 mV.

The composition of Hanks’ solution is shown in Table 2 [44].

The study results were analyzed using VersaStudio (AMETEK Scientific Instruments, Kingston, UK) and EC-Lab^®^ software V11.10 (Biologic Science Instruments, Seyssinet-Pariset, France).

## 3. Results and Discussion

### 3.1. Plasma Electrolytic Oxidation (PEO)

Figure 2 shows the electrical parameters for the PEO process.

The PEO process can be divided into three main stages [45,46], as shown in Figure 2a,b. At the beginning of the surface treatment, the voltage is linearly increased up to a breakdown voltage of 45 V. The current efficiency is almost at the maximum in accordance with Faraday’s law and reaches a value of 0.7 A (0.045 A/cm^2^). The process of forming an amorphous oxide film on the processed sample lasts 3 to 5 min. At this stage, active gas evolution occurs on the sample surface. Upon reaching 45–48 volts, the process passes to the second stage, which lasts approximately 10 min and is characterized by low-power microarc breakdowns. In addition, the resistance to charge transfer increases, and the current decreases from 0.7 to 0.5 A (Figure 2b). This can be addressed by the transformation of the amorphous structure into a crystalline structure and the further growth of the coating.

In the third stage, the power of local breakdowns is increased, and their number is decreased, which corresponds to a compaction of the formed coating and an increase in its dielectric properties.

Depending on the conductivity of the oxide film and the properties of the semiconductor, the resulting two-junction metal-oxide-electrolyte system of the Schottky type can be shifted in the forward or reverse direction [47,48]. When the junction is forward-biased, the PEO process has a vapor gaseous envelope (VGE) surrounding the working electrode. With a reverse-biased junction, the PEO at the anode is characterized by forming an immobile oxide film, which accounts for most of the voltage drop. The most significant voltage drop occurs at the VGE, a non-stationary object with a strong electric field that contributes to the occurrence of plasma discharge in gaseous oxygen media. Micro discharges appear on its surface when the field reaches the breakdown point in this thin film. These phenomena distinguish PEO from other electrolytic processes and justify the term “plasma”.

The nature of the process itself can be described by the waveform plot shown in Figure 2c. During the time period τ_off_, when the voltage and current are not applied, Ti ions can produce a thin amorphous coating on the substrate surface. When an anode current and voltage (τ_on_) are applied, dielectric breakdown occurs, leading to a micro discharge, which forms an oxide-ceramic coating on the metal surface.

It should be noted that the above stages of coating formation in molten salts have similar patterns in aqueous electrolytes. However, they proceed at lower voltages and amount to approximately 50 V at the final stage of the process compared to 175 V in aqueous electrolytes. This difference may occur due to the physical-chemical properties of the molten electrolyte, namely, a higher temperature of the electrolyte and conductivity that is an order of magnitude higher.

### 3.2. Morphology and Chemical Analysis

The coating morphology and cross-section obtained by SEM for the first step of the PEO process are illustrated in Figure 3.

The study of the morphology of the coated surface reveals several changes associated with various electrical parameters used in the PEO process, particularly those associated with the magnitude and size of the coating porosity. The pores on the oxide surface are formed by various types of discharges classified by Hussein et al. [49] in an aqueous electrolyte and by Sobolev et al. [50] in molten salts (Figure 3a,d,g). The formation of these discharges directly depends on their power density, and, consequently, on the morphology of the coating. The high-energy discharge creates micron-sized pores on the surface. With a decrease in the power density, and, as a result, a decrease in the electric field strength, the power of the discharges decreases. The size of the formed pores decreases to submicron sizes in the range of 0.1–0.2 μm. However, one should not forget that, usually, the formed coating has a heterogeneous structure and mixed coating morphology formed by both high- and low-energy breakdowns.

Evaluating the coating morphology of sample S1 (Figure 3a,b) reveals a uniform distribution of micron and submicron surface pores formed by low- and high-energy breakdowns. The cross-section of the sample has a dense, nonporous structure with a coating thickness of 2.4–2.7 μm.

With an increase in the frequency of the current and voltage, both the duration of the micro arc discharge in the anode half-cycle and its single pulse energy decrease, resulting in a decrease in the power of the micro arc discharges. Further SEM study reveals that the surface structure formed for sample S2 (Figure 3d,e) has a smaller number of micron pores and consists of ≈60% submicron pores. Additionally, due to a decrease in the energy of a single pulse, the coating growth rate slightly decreases and has a thickness of 2.0–2.5 μm. With a further increase in the frequency of the current and voltage (sample S3, Figure 3g), micron-sized pores formed by high-energy pulses were not observed. The surface consists of ≈60% submicron pores and ≈40% nanopores (Figure 3g,h). The thickness of the formed coating is decreased by almost 1.5 times compared to sample S1 with a value of 1.2–2.2 μm.

No significant changes in the chemical composition of the analyzed samples were detected, as can be observed in the elemental distribution mapping images (Figure 3b,e,h) and the EDS elemental analysis results, which are shown in Table 3.

Furthermore, the obtained samples were treated by the HTT method for the formation and growth of HA crystals. Precursors were introduced into the coating in the form of a fine powder. The coating morphologies and cross-sections obtained by SEM are illustrated in Figure 4.

After the HTT process, the previously introduced HA precursors were grown in the coating under the influence of temperature and pH [40], resulting in the formation of hexagonal needle-like crystals with sizes of 0.5–1 μm. The morphology, presented earlier in Figure 3, affects the surface filled with HA crystals, and, thereby, leads to the closure of previously formed pores. With this decrease in porosity (Figure 3a,c,e), the number of crystals decreased, and their sizes increased, which is confirmed by the EDS mapping of calcium and phosphorus (Figure 4a,c,e). The EDS chemical analysis data are presented in Table 4.

The presented values in Table 3 and Table 4 indicate a difference in the surface compositions before and after the HTT process. The decrease in the fraction of Ti atoms is associated with the filling of the surface with HA crystals. This behavior is associated with the diffusion, nucleation, and crystallization processes on the sample surface. As described by Liu et al., Ca and P ions diffuse from the inner layer to the surface during HTT [51], resulting in the formation of HA crystals. The optimal Ca/P ratio for HA is 1.67 [44]. Most likely, the content of CaO crystals on the surface of the samples is increased simultaneously with an increase in the Ca/P ratio.

It should also be noted that HTT treatment did not significantly affect the structure of the coating, which has been confirmed by the cross-sections of samples S1–S3 (Figure 3c,f,i) and S1 HTT–S3 HTT (Figure 4b,d,f), respectively.

This study points out that the best surface is obtained using the current frequency of 100 Hz. The frequency increase negatively affects the structure of the oxide coating before and after HTT. With the increase in current frequency and voltage, the formed HA shows a nonstoichiometric composition and contains additional Ca ions, forming other compounds that differ from HA.

### 3.3. Surface Chemistry Analyses

A study of the chemical composition of the surface was performed using XPS analysis of the elements Na, K, O, Ti, Ca, and P. The results obtained for the surface of the S1 HTT sample are presented in Figure 5.

According to the analysis results, O, Ti, Ca, and P were found, associated with titanium oxides and HA. Sodium and potassium from the composition of the electrolyte were not detected.

### 3.4. Phase Analyses

The phase compositions of the coatings were analyzed using the X-ray patterns presented in Figure 6.

Based on Figure 6a, it can be concluded that samples S1–S3 contain two phases of titanium oxide (anatase and rutile) and a low quantity of amorphous addition. Apparently, during the formation of the coating with an increase in the frequency of the current and voltage and a decrease in the power of a single pulse, the temperature in the discharge channel decreases, which contributes to the phase transition from the amorphous state of titanium oxide to the crystalline state [52]. As is known [53], anatase is a lower temperature phase of titanium oxide, and during the PEO process, under the influence of high temperatures, it is converted to rutile. Judging by the graphs shown in Figure 6a, with a decrease in the power of a single breakdown (S1 > S2 > S3), the intensity of the anatase peaks decreases, and, as a consequence, the intensity of the rutile peaks increases.

As shown in Figure 6b, after the HTT treatment, the intensity of the peaks of the rutile phase remains almost unchanged, but the amount of the anatase phase is increased. This dependence is conditioned because rutile, anatase, and an amorphous component has formed during the treatment of PEO, which, upon HTT, is reformed into the lowest temperature and stable crystalline phase of titanium oxide-anatase.

Apatite introduced earlier into the coating under the influence of temperature and pH forms individual HA phases on the surface of the coating in the form of hexagonal needles (Figure 4), which is also confirmed by the EDS and XPS results. Upon further analysis of the X-ray patterns, it can be observed that the intensities of the HA peaks for the three samples (Figure 6b) are of a similar order, and their concentrations in the coating are comparable.

Shin et al. [54] showed that porous coatings of titanium oxide (rutile and anatase) formed on the surface of the implanted material could act as nuclei for the formation of HA in the simulated body fluid, thereby increasing the rate of osseointegration. However, titanium oxides themselves are not a bioactive material but rather only a substrate for the effective growth of HA; therefore, HA compounds were used as a dopant, which can increase the efficiency of osseointegration.

Additional phases consisting of electrolyte components were not found. It is difficult to determine the quantitative phase composition because of the inhomogeneous structure of the coatings, which affects the absorption of X-rays.

The XRD analysis revealed the presence of the following phases present in the coating: Rutile, anatase, amorphous titanium oxide, and crystalline HA. These data are necessary to consciously control the composition by varying the process parameters.

### 3.5. Surface Wettability

The synthesized HA-containing coating was developed to improve the functionality of implants that contact body fluids and merge with human bone tissue. Spijker et al. [55] considered the effect of wetting on the biomaterial interaction, describing the activation of blood proteins, which leads to the formation of blood clots between the surface of the implant and surrounding tissue. To evaluate the effectiveness of the osseointegration process for the surface, a wettability test was performed using contact angle (CA) measurements in Hanks’ solution.

The drop shapes of Hanks’ solution at 37 °C on all samples are shown in Figure 7.

The measured wetting angles are shown in Table 5.

As observed from the values given in Table 5, the CA is significantly decreased after PEO treatment (samples S1–S3) and further decreased after the incorporation of HA crystals (samples S1 HTT–S3 HTT). It can also be observed that the contact angle tends to increase for samples S1 < S2 < S3 with increasing current frequency and voltage during PEO. This trend also persists after hydrothermal treatment S1 HTT < S2 HTT < S3 HTT; however, due to the growth of HA crystals, the physical properties of the surface are changed, and the contact angle is decreased by almost two times compared to samples S1–S3. According to this study, the most hydrophilic coating with the smallest contact angle was determined for sample S1 HTT (CA = 42.7 ± 2.2°). Based on this, the porous structure of the coating and its hydrophilic properties have a positive effect [56] on the rate of osseointegration and the quality of splicing, as it has a more complex relief and a high specific surface.

Since the developed surface is chemically inhomogeneous, the CA behavior can be described by the Cassie–Baxter model [57,58]:(1)$$\cos\theta_{c}=f\left(\cos\theta_{e}+1\right)-1$$
where *θ_c_* is the static CA on a rough, porous surface, *θ_e_* is the CA on a smooth surface of the same chemical nature, and *f* is defined as the fraction of the solid surface wetted by the liquid phase. As follows from this equation, at high values of *f* and low values of *θ_e_*, the formed surface is highly hydrophilic due to low CA.

### 3.6. Electrochemical Measurements

The corrosion behavior of the untreated alloy (S0) and coating samples after PEO (S1–S3) and HTT treatment (S1 HTT–S3 HTT) were analyzed using PPT curves and EIS in Hanks’ solution.

PPT curves are presented in Figure 8, and their calculated values are given in Table 6.

Based on the results of the potentiodynamic polarization curves, the corrosion potentials (*E_corr_*), corrosion current densities (*i_corr_*), and anodic and cathodic slopes of the linear sections (*β_a_* and *β_c_*) were determined by a graphical method. The polarization resistance (*R_p_*) values were calculated using the Stern-Geary equation [59]:(2)$$R_{p}=\frac{\left(\beta_{a}\times \beta_{c}\right)}{2.3\times i_{corr}\left(\beta_{a}+\beta_{c}\right)}$$

Samples S1–S3 demonstrate an increase in the corrosion current and, as a consequence, a decrease in the polarization resistance in the order S1 > S2 > S3 and is 4.03 × 10^6^, 2.29 × 10^6^, and 2.14 × 10^6^ Ohm∙cm^2^, respectively. This pattern is directly related to the frequencies of current and voltage used in the formation of the coating, i.e., with an increase in the frequency of current and voltage, the power of a single breakdown decreases, which does not allow for a uniform oxide structure distributed over the entire area of the treated surface. The evolution of this phenomenon shows in Figure 3.

After the HTT treatment, the regularity of the decrease in the polarization resistance in a series of samples (S1-HTT > S2-HTT > S3-HTT = 4.83 × 10^6^ > 2.98 × 10^6^ > 2.36 × 10^6^ Ohm·cm^2^) remains. However, it has higher values of polarization resistance by 10–15% compared with samples processed in the PEO process. The persistence of this pattern can be explained by the effect of HTT treatment, which, as a result of high temperatures and pressure, transforms the amorphous phase of titanium oxide after the PEO process into a crystalline phase, and also accelerates the formation of HA crystals, which encapsulate coating defects, thereby increasing its corrosion resistance.

As noted earlier (Section 3.3), HTT treatment does not have a significant effect on the cross-section of the samples and, therefore, on the internal structure, which is also confirmed by the analysis of the shape of PPT curves (before and after the HTT process) in semilogarithmic coordinates and its slopes of their linear sections presented in Figure 8. The untreated alloy’s comparative performance has lower polarization resistance values and can be more easily attacked in Hanks’ solution.

The corrosion properties of the studied samples were also evaluated using EIS analysis. EIS analysis also makes it possible to study the influence of the structural features of the samples on their electrochemical behavior in Hanks’ solution. The most used and physically justified equivalent electrical circuits (EEC) for the corrosion process are shown in Figure 9. Figure 9a presents the EEC that simulates the corrosion behavior of an S0 sample with a single “natural” layer, and Figure 9b shows that of a bilayered oxide coating S1–S3 and S1 HTT–S3 HTT, the morphology and cross-section of which are presented in Figure 3 and Figure 4. Electrolyte/electrode interfaces consist of the electrolyte/porous layer and electrolyte/nonporous layer. The designations R_s_, R_1_, CPE_1_, R_2_, and CPE_2_ describe the ECC main elements for this boundary. The R_s_ is the electrolyte resistance, which equals 37 ± 2 Ohm. The series-connected R_1_ is the electrical resistance of the electrolyte in the pores, and CPE_1_ is the geometric capacitance of the porous layer. The R_2_-CPE_2_ has described the process of charge transfer at the nonporous layer/electrolyte interface and represents the charge transfer resistance R_2_ and electric double-layer capacitance CPE_2_ [60].

Based on the coating’s morphology, cross-section, and phase analyses, two equivalent circuits were chosen [61,62], as shown in Figure 9.

Considering the nonideal behavior of the simulated interface, the capacitance can be represented by a constant phase element-CPE, which is described by the following equation [63]:(3)$$Z_{\mathrm{CPE}}=\frac{1}{Y_{0}{\left(j\omega \right)}^{n}}$$
where Z_CPE_ is the impedance of the CPE, Y_0_ is the admittance of an ideal capacitance, j is the imaginary unit (√−1), ω is the angular frequency (2πf), and n is a dimensionless parameter that varies between 0 and 1. At n = 0, the CPE behaves as a resistor, and at n = 1, the CPE behaves as an ideal capacitor.

Figure 10 and Figure 11 present the Nyquist and Bode plots for the S1–S3 and S1 HTT–S3HTT samples, respectively. The calculated values are presented in Table 7.

The fitting of the Nyquist plots reveals a significant difference in corrosion properties between S0 and S1–S3, S1 HTT–S3 HTT samples. Sample S0 has characterized by one capacitive arch (Figure 10 and Figure 11a,b). This capacitive arch can be represented as the charge transfer resistance R_2_ and the capacitance of the electric double layer CPE_2_. The Bode plot has a mid-frequency band with a minimum at approximately −80° (Figure 10 and Figure 11c). The impedance module at low frequencies |Z| _f → 0 Hz_ = 1.49 × 10^6^ Ohm·cm^2^. Comparing the data obtained, the impedance spectrum of this sample can be represented by the equivalent electrical circuit shown in Figure 9a.

From the Nyquist plot for the S1–S3 and S1 HTT–S3 HTT samples, two capacitive arches are found: The first (in the high-frequency range) characterizes the geometric capacitance of the porous layer (CPE_1_) and the electrolyte resistance in the pores (R_1_) [64]. The second arch occurs in the mid- and low-frequency ranges (CPE_2_), which can be attributed to the capacitance of the electrical double layer at the interface between the dense oxide layer and metal substrate and the charge transfer resistance R_2_ [65]. It can be observed from the frequency dependence on the phase angle that there are two characteristic bands (Figure 10c): One is in the high-frequency region, and the second is in the mid-frequency region with 55–70° extrema (Figure 10 and Figure 11c) for both bands. The change in the phase angle (Figure 10 and Figure 11c) is associated with a decrease in the capacitive component of the S1–S3 and S1 HTT–S3 HTT samples, which directly depends on the change in morphology, surface porosity, and density of the resulting coating (Figure 3 and Figure 4).

The EIS data presented in Figure 10a,b in the form of Nyquist and Bode diagrams indicate a positive effect of the formed PEO layer on the protective properties of the samples. Impedance modulus values measured at low frequencies |Z| _f = 0.001 Hz_ for the PEO coating are at least 1.3 times higher than for S0 (Figure 10 and Figure 11, Table 7).

The comparative characteristic of the S1–S3 samples impedance modules has one order of magnitude in the range (2.09–3.51) × 10^6^ Ohm·cm^2^. The main resistance to charge transfer is concentrated on the defects of the nonporous oxide layer, which is a barrier layer since the porous surface layer is characterized by resistance values in the range (0.22–5.21) × 10^5^ Ohm·cm^2^, which are almost an order of magnitude lower than the resistance of the nonporous layer. Thus, the coating formation significantly reduces the possibility of charge transfer at the electrolyte/electrode (porous + nonporous layer) interface, decreasing the rate of corrosion processes.

Table 7 shows the calculated parameters of the electrical circuit elements for the corresponding EEC (Figure 9). Based on the analysis of the presented data (Table 7), it can be concluded that the R_2_ parameter increases and the CPE_2_ values decrease by at least 1.3 for samples with a PEO layer compared to S0. This is a consequence of the significantly greater thickness of the nonporous layer of the PEO coating compared to the thin “natural” oxide layer of the S0 sample. We also note that the values of n_2_ decrease for the samples after being processed by the PEO method (Table 7), which is associated with a reduction in the uniformity of the nonporous layer of the PEO coating compared to the oxide film on the uncoated surface S0.

Based on the analysis of the EIS results, it can be concluded that multistage processing (PEO + HTT) on S1 HTT–S3 HTT samples leads to an increase in the size of the capacitive arch (Figure 11a, Table 7) and an increase in the impedance modulus at low frequencies |Z| _f → 0 Hz_ by a factor of at least 1.2 (Figure 11d, Table 7) compared with samples S1–S3 (PEO). The comparative characteristic of the S1 HTT–S3 HTT and S1–S3 sample impedance modules is one order of magnitude in the range of (2.14–4.03) × 10^6^ Ohm·cm^2^ and (2.41–4.90) × 10^6^ Ohm·cm^2^, respectively. However, as mentioned earlier, the nature of this phenomenon is associated with the encapsulation of the surface pores of the porous layer and internal defects of the non-porous layer during the HTT process, which is also confirmed by an increase in the parameters R_1_ and R_2_ (Table 7) characterizing the electrolyte resistance in the pores and the charge transfer resistance, respectively. Thus, it can be assumed that the decrease in the impedance modulus |Z| _f → 0 Hz_ in the presented samples is associated with a decrease in the available metal surface where the reactions can occur. 

The experimental curves agree with the fitting results (Table 7) since a chi-square value (χ^2^) of approximately 10^−3^ is obtained, which confirms the correctness of the ECC.

Note that the EIS data (Figure 10 and Figure 11, Table 7) are in good agreement with the data obtained by the PPT (Figure 8, Table 6) and also confirm the conclusions made earlier when evaluating the results of SEM regarding the presence of a bilayer structure (Figure 3 and Figure 4) in the PEO coating.

Generally, the corrosion process on the metal/oxide surface proceeds according to the following reactions [66,67]:

The anode reactions:2Ti + 3H_2_O − 6e^−^ → Ti_2_O_3_ + 6H^+^(4)
Ti_2_O_3_ + 3H_2_O − 2e^−^ → 2TiO(OH)_2_ + 2H^+^(5)
Ti + 3H_2_O − 4e^−^ → TiO(OH)_2_ + 4H^+^(6)

With the following chemical dehydration reaction:TiO(OH)_2_ → TiO_2_ + H_2_O(7)

The cathode reaction (oxygen depolarization):O_2_ + 4e^−^ + 2H_2_O → 4OH^−^(8)

When the samples are immersed in Hanks’ solution with a “natural” oxide film (S0) and an oxide coating (S1–S3, S1 HTT–S3 HTT), an interaction between the substrate and electrolyte components occurs at the locations of defects. The anodic process can be divided into several stages with the formation of intermediate reaction products: At the first stage of the corrosion process, oxidation and hydrolysis occur with the formation of metastable titanium suboxides Ti_2_O_3_ (reaction 4). In the second stage, the metastable Ti_2_O_3_ is oxidized and hydrolyzed (5) until the most stable hydroxide complex TiO(OH)_2_ is formed [68], i.e., transformation into the most stable oxidation state Ti^3+^ → Ti^4+^ [69]. The interaction of metal ions can also be represented as a one-stage reaction (6) forming a hydroxide complex. At the final stage of the process, because of dehydration, water molecules are split off with the formation of amorphous TiO_2_ [66]. The cathodic process (8) involves the depolarization of oxygen, usually occurring in electrolytes whose pH is close to neutral.

Previously obtained TiO(OH)_2_ and TiO_2_ interact with electrolyte components according to the adsorption-chemical mechanism [70,71] with the formation of HA occurring via the following reactions:TiO(OH)_2_ + 2H_2_PO_4_^−^ → TiO(H_2_PO_4_)_2_ + 2OH^−^(9)
TiO(OH)_2_ + HPO_4_^2−^ → TiOHPO_4_ + 2OH^−^(10)
3TiO(OH)_2_ + 2PO_4_^3−^ → (TiO)_3_(PO_4_)_2_ + 6OH^−^(11)
3TiO(H_2_PO_4_)_2_ + 10Ca^2+^ + 5H_2_O → 3TiO_2_ + Ca_10_(PO_4_)_6_(OH)_2_ + 20H^+^(12)
6TiOHPO_4_ + 10Ca^2+^ + 8H_2_O → 6TiO_2_ + Ca_10_(PO_4_)_6_(OH)_2_ + 20H^+^(13)
3(TiO)_3_(PO_4_)_2_ + 10Ca^2+^ + 11H_2_O → 9TiO_2_ + Ca_10_(PO_4_)_6_(OH)_2_ + 20H^+^(14)
TiO_2_ + 10Ca^2+^ + 6PO_4_^3-^ + 2H_2_O → TiO_2_ + Ca_10_(PO_4_)_6_(OH)_2_ + 2H^+^(15)

L. Frauchiger et al. [72] detected that the phosphate anions can enter exchange reactions (9–11) with a hydrated surface to form complex titanium phosphates, which react with calcium ions to form HA and TiO_2_. Furthermore, amorphous TiO_2_ can adsorb Ca^2+^ and PO_4_^3−^ ions with HA formation. Therefore, it is logical to assume that the previously introduced apatite precursors have a positive effect on the bioactivity and the rate of osseointegration during the formation of the coating.

The complexity of the object under study, and the variety of chemical and electrochemical processes, both during the coating formation and when exposed to a corrosive environment, led to certain experimental limitations in the performance of the work. In particular, it was impossible to carry out measurements in the low-frequency range when using the EIS method. The results obtained do not yet allow us to conclude the mechanism of the corrosion failure of the coating. This requires special methods to identify the features of individual stages of the corrosion process and their flow regimes.

## 4. Conclusions

This work studied the effects of the current frequency on forming an oxide coating using the PEO method in an electrolyte of molten salts in the range of 100–500 Hz. It was found that an increase in the current frequency leads to a decrease in the instantaneous breakdown power of the dielectric and, as a result, a decrease in the porosity level and a change in the phase composition of the coating. Thus, the share of anatase in the rutile/anatase ratio increases, and the share of the amorphous component increases.

The effect of coating treatment by hydrothermal treatment (HTT) on the morphology and phase composition was studied, and it was found that this type of treatment leads to the transformation of the amorphous phase of titanium into a crystalline one (anatase). As a result of HTT, it is possible to form the HA phase based on previously introduced apatite precursors containing phosphorus and calcium. The results made it possible to explain the coating formation mechanism and identify the optimal electrical parameters necessary to obtain the best corrosion properties.

A contact angle test was carried out and found that the surface’s hydrophilicity decreases with the increased frequency of the PEO process’s current and hydrothermal treatment. The hydrophilicity of the surface improves the wettability of biological fluids and indicates an increase in its bioactivity and, consequently, an acceleration of the osseointegration process.

Electrochemical measurements confirm the conclusions made earlier when evaluating the results of SEM that the presence of bilayer structure (Figure 3 and Figure 4) in the PEO coating. The use of two independent methods, EIS and PPT, made it possible to determine the important electrochemical parameters of the process-corrosion current and potential, as well as polarization resistance. A variant of the mechanism for the formation of the HA structure is proposed, describing the interaction of titanium oxyphosphates of variable valence with calcium ions:6TiOHPO_4_ + 10Ca^2+^ + 8H_2_O → 6TiO_2_ + Ca_10_(PO_4_)_6_(OH)_2_ + 20H^+^
3(TiO)_3_(PO_4_)_2_ + 10Ca^2+^ + 11H_2_O → 9TiO_2_ + Ca_10_(PO_4_)_6_(OH)_2_ + 20H^+^
TiO_2_ + 10Ca^2+^ + 6PO_4_^3-^ + 2H_2_O → TiO_2_ + Ca_10_(PO_4_)_6_(OH)_2_ + 2H^+^

The analysis results testify to the good resistance of the coating in Hanks’ solution, which imitates blood plasma. It also makes it possible to provide a high level of osseointegration due to the formation of a HA structure on the surface.

In the future, the following methods will be used for a deeper study of corrosion processes, namely the elucidation of the kinetics and mechanisms of the process: The disk rotating electrode method, the EIS method with long-term exposure in a corrosive solution of samples, etc. Based on these studies, an analysis of the course of individual reactions will be carried out; a mathematical model will also be built and justified based on experimental results.

## Figures and Tables

**Figure 1 materials-15-07374-f001:**
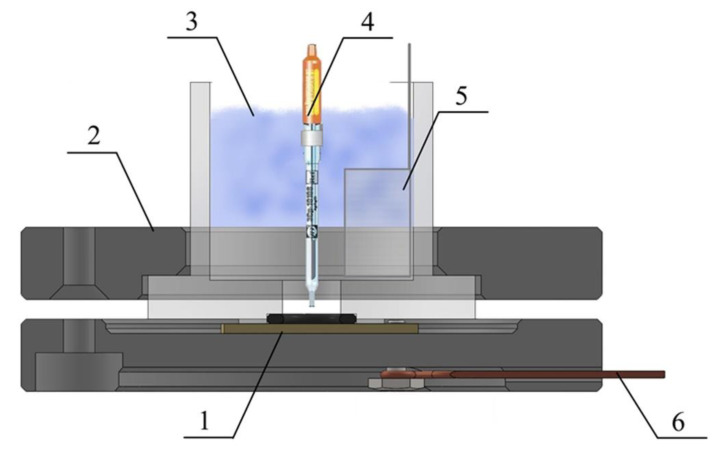
Tree electrodes corrosion cell: 1—working electrode (Sample), 2—corrosion cell, 3—Hanks’ solution (electrolyte), 4—reference electrode (Ag/AgCl_sat_), 5—counter electrode (platinum mesh), 6—current connector.

**Figure 2 materials-15-07374-f002:**
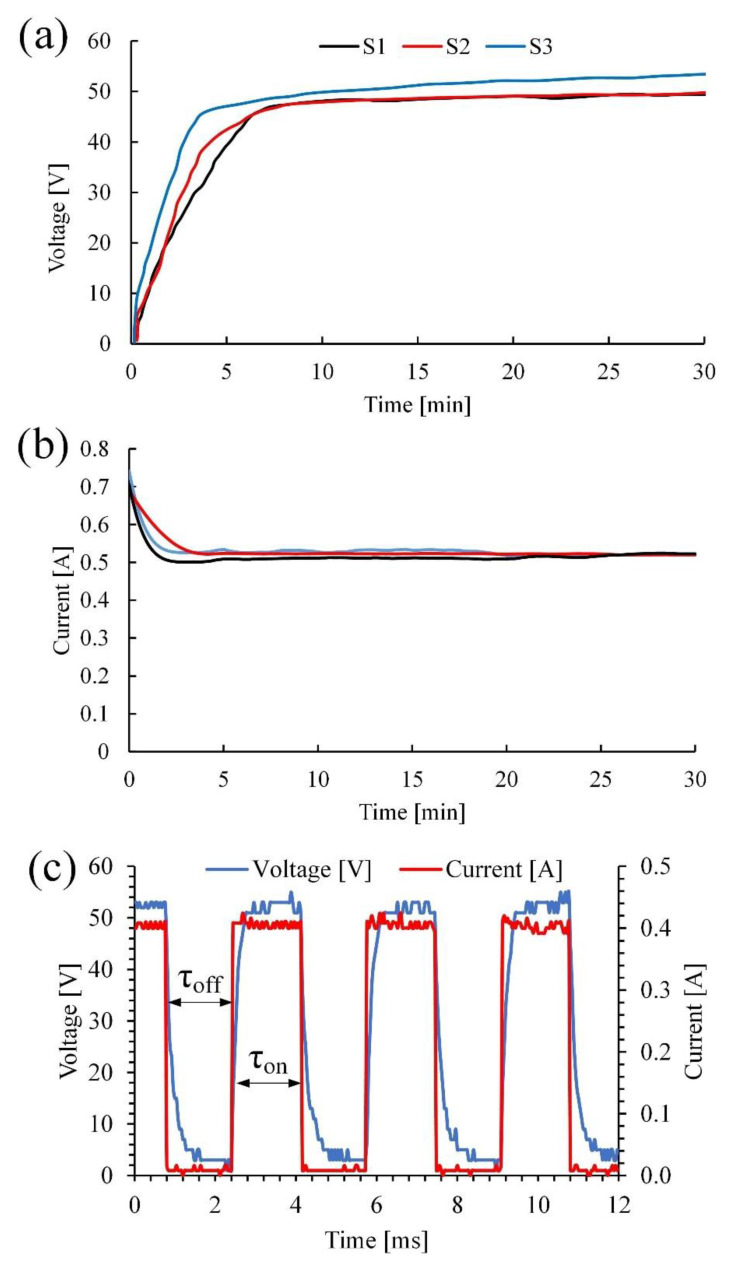
Voltage– and current–time and waveform curves for the PEO surface oxidation process of the Ti-6Al-7Nb alloy. (**a**)Voltage–time plot; (**b**) Current–time plot; (**c**) Wave form plot.

**Figure 3 materials-15-07374-f003:**
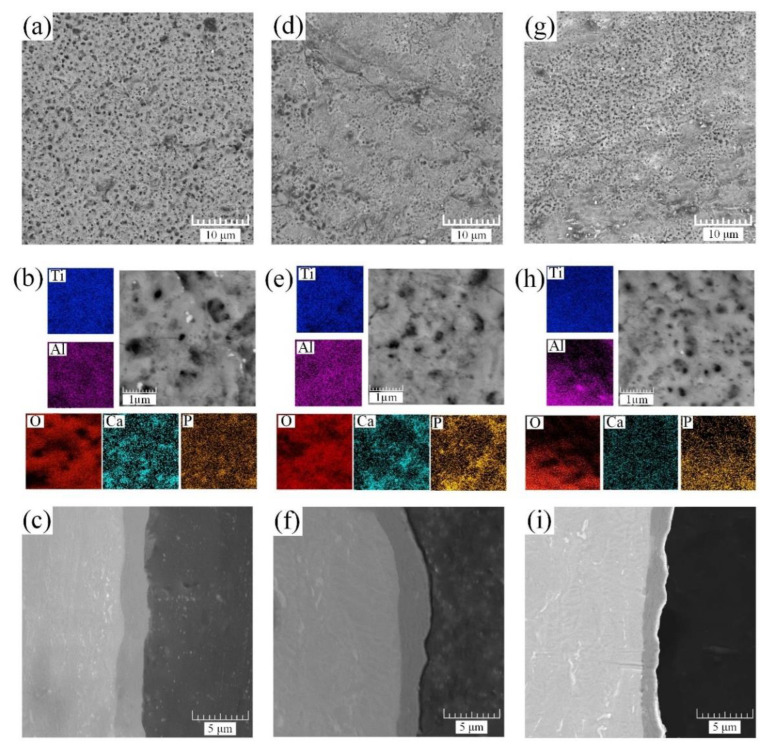
SEM images of the morphology and cross-section of the surfaces of the coated titanium alloys with elemental mapping: (**a**–**c**) S1; (**d**–**f**) S2; (**g**–**i**) S3. Color scale for determining the chemical elements: Blue—Ti; purple—Al; green—Nb; red—O; turquoise—Ca; yellow—P.

**Figure 4 materials-15-07374-f004:**
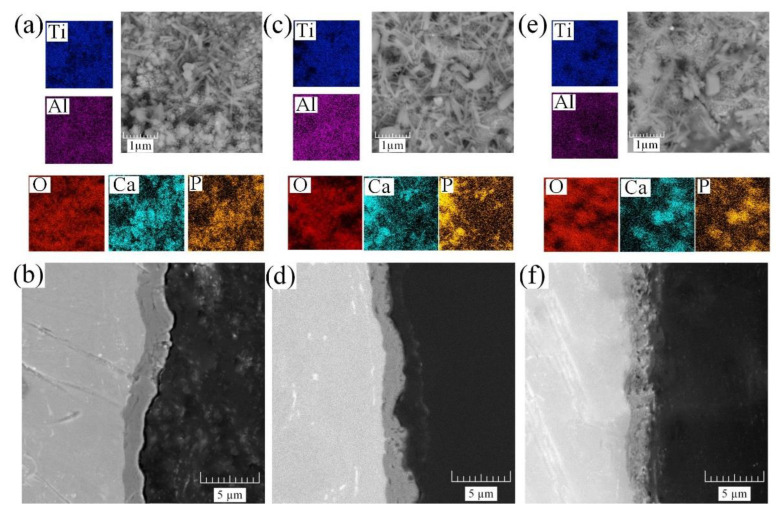
Morphology, EDS analysis, and cross-section of coated samples after the HTT process: (**a**,**b**) S1 HTT; (**c**,**d**) S2 HTT; (**e**,**f**) S3 HTT. Color scale for determining the chemical elements: Blue—Ti; purple—Al; green—Nb; red—O; turquoise—Ca; yellow—P.

**Figure 5 materials-15-07374-f005:**
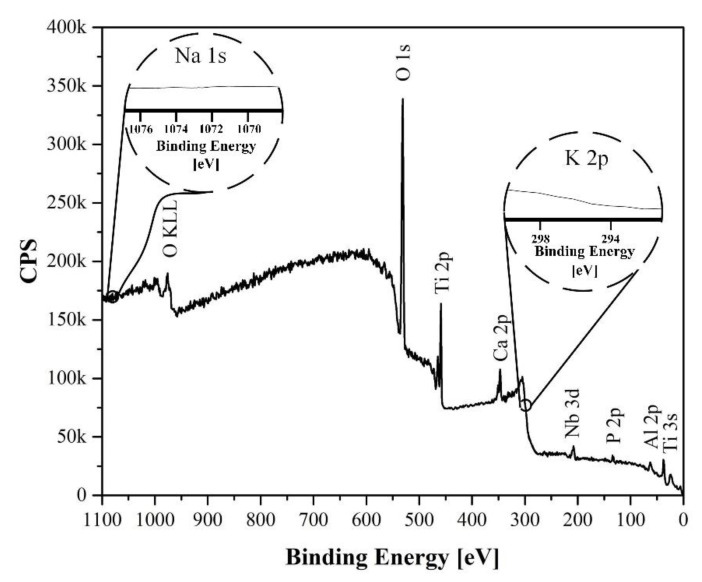
XPS survey spectra for sample S1 HTT.

**Figure 6 materials-15-07374-f006:**
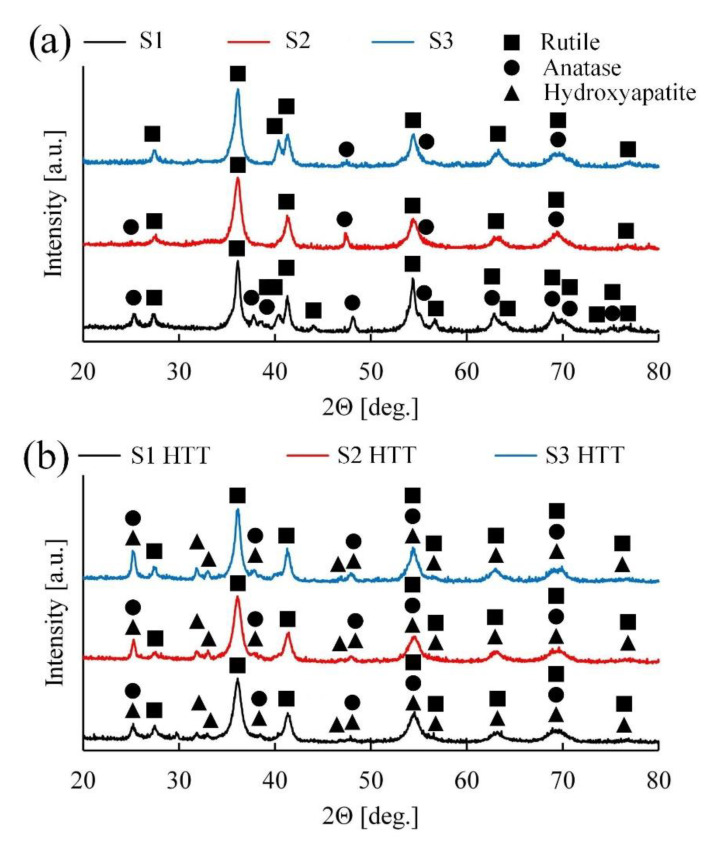
XRD patterns for the coated samples S1–S3 (**a**) and S1 HTT–S3 HTT (**b**).

**Figure 7 materials-15-07374-f007:**
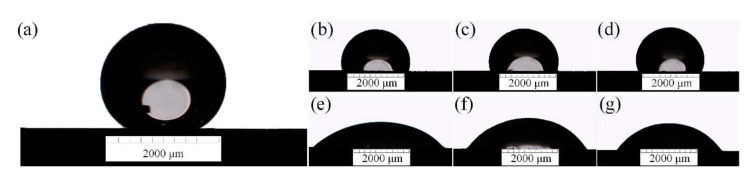
The drop shapes for Hanks’ solution at 37 °C on the sample surface: (**a**) S0; (**b**) S1; (**c**) S2; (**d**) S3; (**e**) S1 HTT; (**f**) S2 HTT; (**g**) S3 HTT.

**Figure 8 materials-15-07374-f008:**
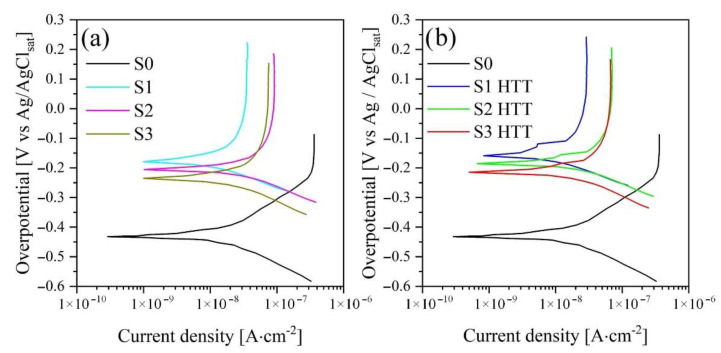
PPT curves were obtained in Hanks’ solution for the following samples: (**a**) S0–S3; (**b**) S0 and S1 HTT–S3 HTT.

**Figure 9 materials-15-07374-f009:**
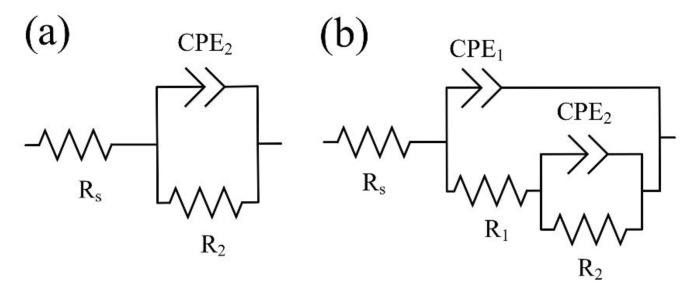
An equivalent electrical circuit was used to fit the EIS spectra for the S0 (**a**) and S1–S3, S1 HTT–S3 HTT (**b**) samples.

**Figure 10 materials-15-07374-f010:**
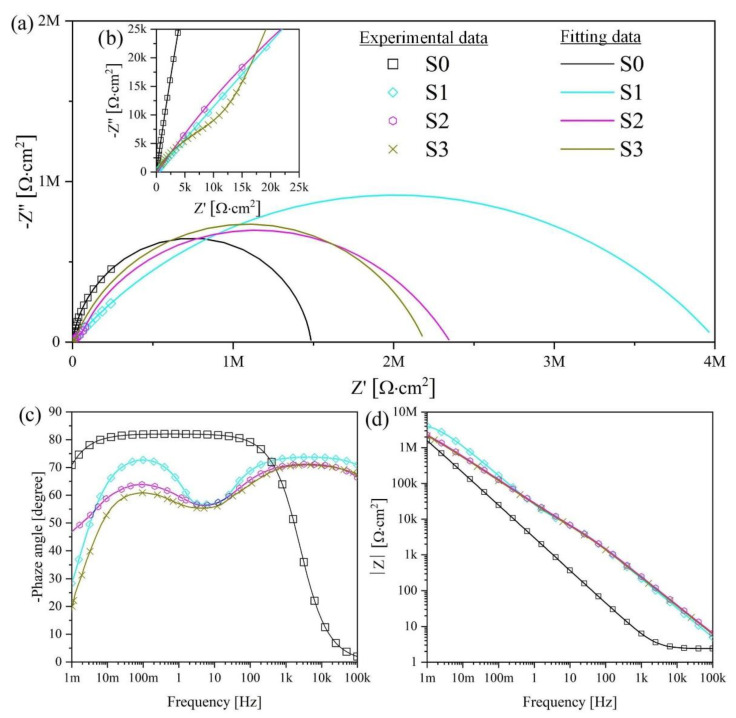
The Nyquist and Bode plots for the S0 and S1–S3 samples obtained in Hanks’ solution: (**a**) Curves for the full frequency range of the impedance; (**b**) low-frequency impedance curves; (**c**) Bode-phase plots; (**d**) Bode impedance plots. Symbols refer to experimental values, and solid lines refer to fitted data.

**Figure 11 materials-15-07374-f011:**
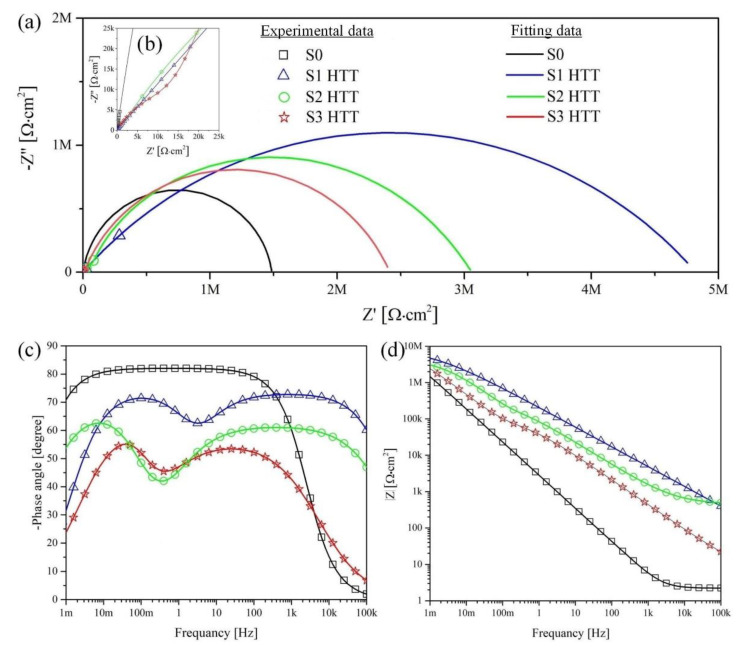
The Nyquist and Bode plots for the S0 and S1 HTT–S3 HTT samples obtained in Hanks’ solution: (**a**) Curves for the full frequency range of the impedance; (**b**) low-frequency impedance curves; (**c**) Bode-phase plots; (**d**) Bode impedance plots. Symbols refer to experimental values, and solid lines refer to fitted data.

**Table 1 materials-15-07374-t001:** Chemical composition of the Ti-6Al-7Nb alloys.

Chemical Element	Fe	Ta	Nb	Al	Ti
Weight %	<0.25	<0.5	6.5–7.5	5.5–6.5	Balance

**Table 2 materials-15-07374-t002:** Composition of Hanks’ solution.

Components	Concentration (mM)
KCl	5.333
KH_2_PO_4_	0.441
NaHCO_3_	4.167
NaCl	137.9
Na_2_HPO_4_	0.338
CaCl_2_	1.261
MgCl_2_·6H_2_O	0.493
MgSO_4_·7H_2_O	0.407
D-glucose	5.556

**Table 3 materials-15-07374-t003:** EDS elemental analysis data for samples S1–S3.

Samples	O, at. %	Ti, at. %	Al, at. %	Nb, at. %	Ca, at. %	P, at. %
S1	69.2	24.6	2.8	1.1	1.0	1.3
S2	68.9	25.1	2.8	1.1	0.9	1.2
S3	69.5	24.8	2.7	1.2	0.8	1.0

**Table 4 materials-15-07374-t004:** EDS elemental analysis data for samples S1 HTT–S3 HTT.

Sample	O, at. %	Ti, at. %	Al, at. %	Nb, at. %	Ca, at. %	P, at. %	Ca/P
S1 HTT	69.5	15.2	2.6	1.0	7.5	4.2	1.8
S2 HTT	68.9	15.8	2.6	1.1	7.6	4.0	1.9
S3 HTT	69.6	16.2	2.7	0.9	7.3	3.3	2.2

**Table 5 materials-15-07374-t005:** Measured contact angles.

Samples	S0	S1	S2	S3	S1 HTT	S2 HTT	S3 HTT
CA [°]	135.2 ± 0.7	96.0 ± 1.4	99.6 ± 0.2	104.7 ± 0.4	42.7 ± 2.2	47.9 ± 1.3	49.3 ± 1.5

**Table 6 materials-15-07374-t006:** The main electrochemical parameters were obtained by analyzing the PPT curves in Hanks’ solution.

Samples	*i_corr_* × 10^−9^ [A∙cm^−2^]	*E_corr_* vs. Ag/AgCl_sat_ [V]	*β_a_* [V∙dec^−1^]	−*β_c_* [V∙dec^−1^]	*R_p_* × 10^6^ [Ω∙cm^2^]
S0	20.21 ± 0.15	−0.434 ± 0.003	0.161 ± 0.006	0.133 ± 0.002	1.57 ± 0.01
S1	10.11± 0.07	−0.178 ± 0.003	0.320 ± 0.003	0.136 ± 0.002	4.13 ± 0.02
S2	18.20 ± 0.10	−0.196 ± 0.003	0.322 ± 0.006	0.136 ± 0.004	2.23 ± 0.02
S3	19.16 ± 0.14	−0.223 ± 0.003	0.333 ± 0.004	0.133 ± 0.002	2.04 ± 0.02
S1 HTT	8.43 ± 0.06	−0.166 ± 0.002	0.316 ± 0.005	0.137 ± 0.003	4.83 ± 0.02
S2 HTT	14.06 ± 0.10	−0.182 ± 0.003	0.324 ± 0.008	0.134 ± 0.002	2.98 ± 0.02
S3 HTT	17.42 ± 0.14	−0.201 ± 0.002	0.331 ± 0.005	0.132 ± 0.003	2.36 ± 0.02

**Table 7 materials-15-07374-t007:** The electrical parameters used for fitting the EIS data in Hanks’ solution.

Samples	S0	S1	S2	S3	S1 HTT	S2 HTT	S3 HTT
CPE1 × 10^−6^ [F·cm^−2^·s^n−1^]	–	4.27	2.45	1.64	6.73	2.07	1.49
n_1_	–	0.62	0.64	0.68	0.64	0.67	0.69
R_1_ × 10^5^ [Ω·cm^2^]	–	5.21	2.11	0.22	6.25	2.53	0.27
CPE_2_ × 10^−7^ [F·cm^−2^·s^n−1^]	0.32	1.72	2.31	3.88	1.64	1.99	3.42
n_2_	0.92	0.89	0.91	0.79	0.90	0.87	0.83
R_2_ × 10^6^ [Ω·cm^2^]	1.49	3.51	2.09	2.12	4.21	2.81	2.38
Z_f→0 Hz_ × 10^6^ [Ω·cm^2^]	1.49	4.03	2.29	2.14	4.9	3.06	2.41
χ^2^ × 10^−3^	1.01	1.22	1.04	1.14	1.12	1.08	1.10

There may be a 1–2% variance in the results of all specimens.

## Data Availability

All the data supporting the findings of this study are available within the article.

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
