# Peer review of "Hydroxyapatite Coating on Ti-6Al-7Nb Alloy by Plasma Electrolytic Oxidation in Salt-Based Electrolyte"

_materials, 2022, doi:10.3390/ma15207374_

Round 1

Reviewer 1 Report

The manuscript is on bio-ceramic coating formation on Ti-6Al-7Nb ally by PEO technique in salt-based electrolyte. Manuscript is formulated well, and results are interesting. Below are my comments.

1.       In Figure 2 and Figure 3, EDS maps are not very visible. Please enlarge the EDS maps.

2.       In Figure 9a and Figure 10a, experimental Nyquist data (data point represented as symbols) are not there after resistance (Z’) ~0.5M. Please check the equivalent circuit diagram and fitting.

Author Response

Dear Reviewer,

The authors are very grateful for your valuable comments, which improve the qualitative analysis of the results and give a better understanding of the issue's essence in the present work.

The manuscript is on bio-ceramic coating formation on Ti-6Al-7Nb ally by PEO technique in salt-based electrolyte. Manuscript is formulated well, and results are interesting. Below are my comments.

  1. In Figure 2 and Figure 3, EDS maps are not very visible. Please enlarge the EDS maps.

Thanks for the valuable remark. Elemental mapping pictures have been enlarged.

  1. In Figure 9a and Figure 10a, experimental Nyquist data (data point represented as symbols) are not there after resistance (Z’) ~0.5M. Please check the equivalent circuit diagram and fitting.

Thanks for the valuable remark.

For the EIS method, scanning generally uses a low-frequency range. However, due to the complex structure of the coating, it is not possible to measure in the entire frequency range (100 kHz - 1 μHz). Therefore, we substantiated the chosen equivalent scheme describing the features of the coverage structure, where the indicator of adequacy is the approximation accuracy χ2. In our work, χ2≈ 10-3 corresponds to the high approximation accuracy and the adequacy of the equivalent circuit.

Author Response

Dear Reviewer,

The authors are very grateful for your valuable comments, which improve the qualitative analysis of the results and give a better understanding of the issue's essence in the present work.

 Revisions:

  • line 15: the word “method” should be “methods”. The singular countable noun method follows the quantifier “one of…”, which requires a plural noun. Consider using a plural noun or a different quantifier.

Thanks for the valuable remark. Replaced to ’’methods’’.

  • Line 18/105: do you use apatite or hydroxyapatite nanoparticles?

We use apatite nanoparticles as a precursor, but they turn into hydroxyapatite form after hydrothermal treatment.

  • Line 94/109: the 2 square cm should be a superscript

Corrected

  • Line 104: please correct all subscripts in chemical formulas

The submitted text has been checked and corrected.

  • Table 2: the nomenclature for glucose is D-glucose, in capital letter

Corrected

  • Figure 1: please enlarge the images inside figure 1, perhaps arranging them on several lines.

Corrected

  • Line 350: there is an extra "f" in the text.

Corrected

  • The figures in the text must be named "Fig." or "Figures". Choose one, and make the text homogeneous.

Corrected

  • In reaction (9), you should correct the formula of the anion dihydrogen phosphate. The 4 is a non-superscript subscript.

Corrected

  • It would be best if you corrected the stoichiometry of the reaction (10).

Corrected

  • In reaction (12) and (13), the hydrogens and oxygen are not balanced.

Corrected

  • In the reaction (15) between titanium dioxide and hydroxyapatite, there is a "+".

Corrected

  • Do the reactions (9-11) all take place? Considering the pH of the solution and the presence of the bisphosphate and dihydrogen phosphate anions already present in the Hanks solution, it is possible that only reactions (9) and (10) occur since the phosphate anion is unlikely to form at neutral pH. For the same reason, reactions (14) and (15) are also improbable. Could you add some considerations to support your thesis?

The remark is fair; we cannot give an unambiguous conclusion about the course of specific reactions. At the same time, we considered it possible to assume the occurrence of reactions 9 -11, taking into account the complexity of the system and the implementation of both chemical and electrochemical processes at the boundary. In this case, the formation of titanium cations of various valences is possible, resulting in the formation of phosphate anions according to reaction 11. As a result, reactions 14 and 15 will be possible.

  • In my opinion, the conclusions should not be written with a bulleted list.

Conclusions revised

Reviewer 3 Report

1.      The abstract should be broadened to give additional quantitative results.

2.      Please conclude your abstract with a "take-home" message.

3.      Rearrange the keywords in terms of alphabetical order.

4.      Please use lowercase font for each term following MDPI format.

5.      It is encouraged not used abbreviations in the keywords section.

6.      It is unclear whether the author's something new in this work. According to evaluation, several published studies by other researchers in the past adequately explain the issues you made in the present paper. Please be careful to highlight in the introduction section anything really innovative in this work.

7.      To underline the study gaps that the newest research tries to fill, it is crucial to explain the merits, novelty, and limits of earlier studies in the introduction.

8.      In the introduction section line 30-53, the authors need to explain the advantage of titanium and its alloy from perspective of biotribology, biomechanics, and biocompability for the reason it is suitable for use in the medical application. is a vital topic that authors must provide in the revised manuscript Additionally, the MDPI's suggested reverence should be taken to substantiate this explanation as follows: Jamari, J.; Ammarullah, M. I.; Santoso, G.; Sugiharto, S.; Supriyono, T.; van der Heide, E. In Silico Contact Pressure of Metal-on-Metal Total Hip Implant with Different Materials Subjected to Gait Loading. Metals (Basel). 2022, 12, 1241. https://doi.org/10.3390/met12081241

9.      To help the reader grasp the study's workflow more easily, the authors could include more visuals to the materials and methods section in the form of figures rather than sticking with the text that now predominates.

10.   It's also important to provide more particular information on tools, such as the manufacturer, the country, and the specification.

11.   Error and tolerance of experimental tools used in this work are important information that needs to be explained in the manuscript. It is would use as a valuable discussion due to different results in the further study by other researcher.

12.   Outcomes must be compared to similar past research.

13.   What is the limitation of the present work? Please include it before the conclusion section.

14.   Express the conclusion in the form of a paragraph rather than in the current form, that is point by point.

15.   Mention further research in the conclusion section.

16.   The reference should be enriched with literature from the last five years. Literature published by MDPI is strongly recommended.

17.   The authors sometimes reduced a paragraph to just one or two phrases across the whole article, which made the explanation difficult to follow. To make a more thorough paragraph, the writers should expand upon their explanation. It is advised to include at least three sentences in a paragraph, one of which should serve as the primary idea and the others as supporting details.

18.   Due to grammatical and linguistic style issues, the authors should proofread the manuscript. For this issue, the authors would utilize the MDPI English editing service.

19.   Please review and confirm that the writers followed the MDPI format exactly, edit the current form, and recheck in addition to the other issues that have been mentioned.

20.   Provide graphical abstract for submission after revision.

Author Response

Dear Reviewer,

The authors are very grateful for your valuable comments, which improve the qualitative analysis of the results and give a better understanding of the issue's essence in the present work

  1. The abstract should be broadened to give additional quantitative results.

Thanks for the comment. We reworked the abstract by adding quantitative characteristics in the amount of 200 words.

  1. Please conclude your abstract with a "take-home" message.

In this work, for the first time, the process of formation of a bioactive coating consisting of titanium oxides and HA was studied by the PEO method in molten salts.

  1. Rearrange the keywords in terms of alphabetical order.

Corrected

  1. Please use lowercase font for each term following MDPI format.

Corrected

  1. It is encouraged not used abbreviations in the keywords section.

Corrected

  1. It is unclear whether the author's something new in this work. According to evaluation, several published studies by other researchers in the past adequately explain the issues you made in the present paper. Please be careful to highlight in the introduction section anything really innovative in this work.
  2. To underline the study gaps that the newest research tries to fill, it is crucial to explain the merits, novelty, and limits of earlier studies in the introduction.

Based on the remarks in paragraphs 6 and 7, we emphasized the PEO method in aqueous shortcomings and the merits of what we propose. Marked in yellow and added to the article.

The bioactive coating fabricated by PEO methods in an aqueous solution usually consists of rutile and anatase crystalline phases with a small amount of HA. However, the introduced HA has a low crystallinity; therefore, the process must be accompanied by an additional hydrothermal treatment [38-40]. Amorphous apatite undergoes a phase transition during hydrothermal treatment, turning into HA. The resulting crystals increase the adsorption-chemical interaction [41] of physiological fluids (calcium and phosphorus ions) with the implant surface, thereby positively affecting the bioactivity and rate of osseointegration.

There are several main drawbacks in the PEO process in an aqueous electrolyte, namely, the use of high currents and voltages, which is decisive, since the dimensions of the processed samples are limited by the power supplied from the power source. In addition, the PEO process in aqueous electrolytes has several disadvantages, such as electrolyte heating, high current density, and the appearance of undesirable impurities in the newly formed coating due to electrolyte decomposition. It should also be noted that the coating has a two-layer through porous structure and contains impurities from the composition of the electrolyte, which worsens the mechanical, anticorrosive, and cytotoxic properties of the coating. In addition, the formed HA crystals have a coarse-grained structure, which negatively affects their mechanical and tribological properties and often leads to their destruction during implantation. Our previous studies have shown that these problems can be solved by conducting a PEO process in molten salt. We studied and analyzed the coating surface and its corrosion properties obtained for Ti-6Al-4V [42] and Al1050 alloys [43] using PEO in molten salt.

In this work, for the first time, the process of formation of a bioactive coating consisting of titanium oxides and HA by the PEO method in molten nitrate salts, and also study the influence of the PEO current and voltage frequency on the morphology, chemical and phase composition, wettability and corrosion properties of fabricated coatings on a Ti-6Al-7Nb alloy. The morphology, phase, and chemical composition of the coatings were examined using scanning electron microscopy (SEM), X-ray diffraction (XRD), and X-ray photoelectron spectroscopy (XPS), respectively. The wettability properties of the surface were investigated using Hank’s solution by the sessile drop method with contact angle measurements. Finally, the corrosion behavior of the developed coatings was studied using PPT and EIS tests in Hank’s solution.

The coating obtained in molten salts does not contain undesirable impurities from the electrolyte composition. It has a two-layer coating structure with an inner dense layer and an outer porous layer with evenly distributed small HA crystals that have high mechanical and corrosion resistance with good bioactive and osseointegration properties.

  1. In the introduction section line 30-53, the authors need to explain the advantage of titanium and its alloy from perspective of biotribology, biomechanics, and biocompability for the reason it is suitable for use in the medical application. is a vital topic that authors must provide in the revised manuscript Additionally, the MDPI's suggested reverence should be taken to substantiate this explanation as follows: Jamari, J.; Ammarullah, M. I.; Santoso, G.; Sugiharto, S.; Supriyono, T.; van der Heide, E. In Silico Contact Pressure of Metal-on-Metal Total Hip Implant with Different Materials Subjected to Gait Loading. Metals (Basel). 2022, 12, 1241. https://doi.org/10.3390/met12081241

Added to manuscript.

  1. To help the reader grasp the study's workflow more easily, the authors could include more visuals to the materials and methods section in the form of figures rather than sticking with the text that now predominates.

Thank you, we have added an electrochemical cell of ’’home made’’ manufacture.

  1. It's also important to provide more particular information on tools, such as the manufacturer, the country, and the specification.

Corrected

  1. Error and tolerance of experimental tools used in this work are important information that needs to be explained in the manuscript. It is would use as a valuable discussion due to different results in the further study by other researcher.

Error and tolerance of experimental tools has been added in manuscript

  1. Outcomes must be compared to similar past research.

Sections 6 and 7 detail the findings and compare them with past studies.

  1. What is the limitation of the present work? Please include it before the conclusion section.

The complexity of the object under study, and the variety of chemical and electrochemical processes, both during the coating formation and when exposed to a corrosive environment, led to certain experimental limitations in the performance of the work. In particular, it was impossible to carry out measurements in the low-frequency range when using the EIS method. The results obtained do not yet allow us to conclude the mechanism of the corrosion failure of the coating. This requires special methods to identify the features of individual stages of the corrosion process and their flow regimes.

  1. Express the conclusion in the form of a paragraph rather than in the current form, that is point by point.

Corrected

  1. Mention further research in the conclusion section.

In the future, the following methods will be used for a deeper study of corrosion processes, namely the elucidation of the process's kinetics and mechanism: the disk rotating electrode method, the EIS method with long-term exposure in a corrosive solution of samples, etc. Based on these studies, an analysis of the course of individual reactions will be carried out; a mathematical model will be built and justified based on experimental results.

  1. The reference should be enriched with literature from the last five years. Literature published by MDPI is strongly recommended.

Corrected and added

  1. The authors sometimes reduced a paragraph to just one or two phrases across the whole article, which made the explanation difficult to follow. To make a more thorough paragraph, the writers should expand upon their explanation. It is advised to include at least three sentences in a paragraph, one of which should serve as the primary idea and the others as supporting details.

Corrected

  1. Due to grammatical and linguistic style issues, the authors should proofread the manuscript. For this issue, the authors would utilize the MDPI English editing service.

When preparing the article, we used the editorial grammar service of the Elsevier system to edit the article. This version of the manuscript, we send you, after additional editing of the English language. If there are other requirements for this release, we are ready to fulfill them.

  1. Please review and confirm that the writers followed the MDPI format exactly, edit the current form, and recheck in addition to the other issues that have been mentioned.

 All necessary adjustments have been made

  1. Provide graphical abstract for submission after revision.

Graphic abstract sent

Reviewer 4 Report

Dear Editor: I would like to express my deep thanks for inviting me to review the manuscript ID: materials-1968467-peer-review-v1

Title:    Bioceramic Coating Formation on Ti-6Al-7Nb Alloy by Plasma Electrolytic Oxidation in Salt-based Electrolyte

Authors: Avital Schwartz, Alexey Kossenko, Michael Zinigrad, Yosef Gofer, Konstantin Borodianskiy and Alexander Sobolev

Comments:

Bioceramic Coating Formation on Ti-6Al-7Nb Alloy by Plasma Electrolytic Oxidation in Salt-based Electrolyte.

Replaced by

Hydroxyapatite Coating on Ti-6Al-7Nb Alloy by Plasma Electrolytic Oxidation in Salt-based Electrolyte

Introduction:

Need to rewrite introduction part.

For example,

“Modern medicine applies metal and polymer materials for the implantation of damaged bones. Most metal implants that are used in medicine are predominantly made of titanium and its alloys. Ti - 6Al - 4V is the most applicable and commercially available alloy for these applications”.

Replaced by

“Metal and polymer artificial implant materials are wieldy employed for repair and reconstruction of damaged bones. Among the metal bioimplants titanium (Ti) and its alloys are mostly used for load bearing parts [1-3]. Particularly, Ti-6Al-4V is the most applicable and commercially available alloy for these applications because of it better mechanical properties and good biocompatibility [4, 5].”

1.       M. Geetha, A.K. Singh, R. Asokamani, A.K. Gogia, Ti based biomaterials, the ultimate choice for orthopaedic implants - A review, Prog. Mater. Sci. 54 (2009) 397-425

2.       A.K. Gain, L. Zhang, M.Z. Quadir, Composites matching the properties of human cortical bones: The design of porous titanium-zirconia (Ti-ZrO2) nanocomposites using polymethyl methacrylate powders, Mater. Sci. and Eng. A 662, (2016) 258-267.

3.       M. Semlitsch, Titanium alloys for hip joint replacements, Clinical Mater. 2 (1987) l-l3.

4.       A.K. Gain, L. Zhang, S. Lim,Tribological behavior of Ti-6Al-4V alloy: subsurface structure, damage mechanism and mechanical properties, Wear 464, (2021) 203551.

5.       S. Liu, Y. C. Shin, Additive manufacturing of Ti6Al4V alloy: A review, Mater. Des. 164 (2019) 107552.

Please explain in detail the objectives and novelty of this work.

Materials and Methods

Explain in detail the characterization section

Results and discussion:

1.              Figure 1 quality is very poor, provide better quality images.

2.              In Figure 2. It is better to provide EDS point analysis spectra at different locations instead of mapping.

3.              Based on Fig. 5a, it can be concluded that samples S1-S3 contain two phases of titanium oxide (anatase and rutile)” it is clear that Ti phase was oxidized during processing of coating, is it influences on biocompatibility of Ti-alloy explain in detail.

4.              Need to provide the biocompatibility analysis data.

Conclusion part:

Please rewrite the conclusion part and concise it.

RECOMMENDATION

After reviewing the enclosed manuscript for “Materials”, the present manuscript contains some kinds of scientific analysis but it is mandatory required to modify according to the preceding remarks. So, the manuscript can be publication after major revision.

Author Response

The authors are very grateful for your valuable comments, which improve the qualitative analysis of the results and give a better understanding of the issue's essence in the present work

Comments:

Bioceramic Coating Formation on Ti-6Al-7Nb Alloy by Plasma Electrolytic Oxidation in Salt-based Electrolyte.

Replaced by

Hydroxyapatite Coating on Ti-6Al-7Nb Alloy by Plasma Electrolytic Oxidation in Salt-based Electrolyte

The title of the article has been changed on your recommendation.

Introduction:

Need to rewrite introduction part. We changed the introduction and marked it in yellow.

For example,

“Modern medicine applies metal and polymer materials for the implantation of damaged bones. Most metal implants that are used in medicine are predominantly made of titanium and its alloys. Ti - 6Al - 4V is the most applicable and commercially available alloy for these applications”.

Replaced by

“Metal and polymer artificial implant materials are wieldy employed for repair and reconstruction of damaged bones. Among the metal bioimplants titanium (Ti) and its alloys are mostly used for load bearing parts [1-3]. Particularly, Ti-6Al-4V is the most applicable and commercially available alloy for these applications because of it better mechanical properties and good biocompatibility [4, 5].”

  1. M. Geetha, A.K. Singh, R. Asokamani, A.K. Gogia, Ti based biomaterials, the ultimate choice for orthopaedic implants - A review, Prog. Mater. Sci. 54 (2009) 397-425
  2. A.K. Gain, L. Zhang, M.Z. Quadir, Composites matching the properties of human cortical bones: The design of porous titanium-zirconia (Ti-ZrO2) nanocomposites using polymethyl methacrylate powders, Mater. Sci. and Eng. A 662, (2016) 258-267.
  3. M. Semlitsch, Titanium alloys for hip joint replacements, Clinical Mater. 2 (1987) l-l3.
  4. A.K. Gain, L. Zhang, S. Lim,Tribological behavior of Ti-6Al-4V alloy: subsurface structure, damage mechanism and mechanical properties, Wear 464, (2021) 203551.
  5. S. Liu, Y. C. Shin, Additive manufacturing of Ti6Al4V alloy: A review, Mater. Des. 164 (2019) 107552.

Please explain in detail the objectives and novelty of this work.

The novelty of this work is added to the introduction and its goals are clarified.

There are several main drawbacks in the PEO process in an aqueous electrolyte, namely, the use of high currents and voltages, which is decisive, since the dimensions of the processed samples are limited by the power supplied from the power source. In addition, the PEO process in aqueous electrolytes has several disadvantages, such as electrolyte heating, high current density, and the appearance of undesirable impurities in the newly formed coating due to electrolyte decomposition. It should also be noted that the coating has a two-layer through porous structure and contains impurities from the composition of the electrolyte, which worsens the mechanical, anticorrosive, and cytotoxic properties of the coating. In addition, the formed HA crystals have a coarse-grained structure, which negatively affects their mechanical and tribological properties and often leads to their destruction during implantation. Our previous studies have shown that these problems can be solved by conducting a PEO process in molten salt. We studied and analyzed the coating surface and its corrosion properties obtained for Ti-6Al-4V [42] and Al1050 alloys [43] using PEO in molten salt.

In this work, for the first time, the process of formation of a bioactive coating consisting of titanium oxides and HA by the PEO method in molten nitrate salts, and also study the influence of the PEO current and voltage frequency on the morphology, chemical and phase composition, wettability and corrosion properties of fabricated coatings on a Ti-6Al-7Nb alloy. The morphology, phase, and chemical composition of the coatings were examined using scanning electron microscopy (SEM), X-ray diffraction (XRD), and X-ray photoelectron spectroscopy (XPS), respectively. The wettability properties of the surface were investigated using Hank’s solution by the sessile drop method with contact angle measurements. Finally, the corrosion behavior of the developed coatings was studied using PPT and EIS tests in Hank’s solution.

The coating obtained in molten salts does not contain undesirable impurities from the electrolyte composition. It has a two-layer coating structure with an inner dense layer and an outer porous layer with evenly distributed small HA crystals that have high mechanical and corrosion resistance with good bioactive and osseointegration properties.

Materials and Methods

Explain in detail the characterization section

Corrected, added and marked in yellow

Results and discussion:

  1. Figure 1 quality is very poor, provide better quality images.

Corrected

  1. In Figure 2. It is better to provide EDS point analysis spectra at different locations instead of mapping.

In this analysis, we were interested in the distribution of elements over the surface. Given the relative accuracy of the EDS, it seems important to consider the distribution of elements over the entire surface and not pointwise.

  1. “Based on Fig. 5a, it can be concluded that samples S1-S3 contain two phases of titanium oxide (anatase and rutile)” it is clear that Ti phase was oxidized during processing of coating, is it influences on biocompatibility of Ti-alloy explain in detail.

Shin et al. [55] showed that porous coatings of titanium oxide (rutile and anatase) formed on the surface of the implanted material could act as nuclei for the formation of HA in the simulated body fluid, thereby increasing the rate of osseointegration. However, titanium oxides themselves are not a bioactive material but only a substrate for the effective growth of HA; therefore, HA compounds were used as a dopant, which can increase the efficiency of osseointegration.

  1. Need to provide the biocompatibility analysis data.

Conclusion part:

Please rewrite the conclusion part and concise it.

Corrected

Round 2

Reviewer 3 Report

I have no any further comments.

Reviewer 4 Report

Authors addressed all the reviewers comments in the revised manuscript.